# Towards Subgraph Isomorphism Counting with Graph Kernels

## Abstract

Subgraph isomorphism counting is known to be #P-complete, requiring exponential time to find an accurate solution. Recent advancements in representation learning have shown promise in representing substructures and approximating solutions. Graph kernels, which implicitly capture the correlations among substructures in diverse graphs, have demonstrated significant discriminative power in graph classification. We, therefore, explore their potential in counting subgraph isomorphisms and further enhance kernel capabilities through various variants, including polynomial and Gaussian kernels. Through comprehensive analysis, we improve the graph kernels by incorporating neighborhood information. Finally, we present the results of extensive experiments to demonstrate the effectiveness of the enhanced graph kernels and discuss promising directions for future research.

## 1 Introduction

The objective of subgraph isomorphism counting is to determine the number of subgraphs in a given graph that match a specific *pattern* graph, i.e., that are isomorphic to it. This technique is highly valuable in knowledge discovery and data mining applications, such as identifying protein interactions in bioinformatics (Milo et al., 2002; Alon et al., 2008). Moreover, it is beneficial for analyzing heterogeneous information networks (HINs), including knowledge graphs (Shen et al., 2019), online social networks (Kuramochi & Karypis, 2004), and recommendation systems (Huang et al., 2016; Zhao et al., 2017). The diverse range of types within the HIN schema offers meaningful semantics, making subgraph counting a valuable component in various types of queries.

Numerous backtracking algorithms and indexing-based approaches have been proposed to tackle the challenges of subgraph isomorphisms (Ullmann, 1976; Cordella et al., 2004b; He & Singh, 2008; Han et al., 2013; Carletti et al., 2018; Klein et al., 2011). However, previous research on similar tasks often focuses on specific constraints, with limited discussions on general patterns in heterogeneous graphs. Due to the NP-hard nature of subgraph isomorphisms, researchers have also explored efficient approximations of the number of subgraph isomorphisms instead of exact counts, using techniques such as sampling (Jha et al., 2015) and color coding (Zhao et al., 2012). While these approximate solutions are relatively efficient, generalizing them to heterogeneous settings is challenging, especially considering the high memory consumption and dynamic programming complexity in heuristic rules.

Graph-based learning has recently gained significant interest, and graph kernel methods have been extensively applied in machine learning for graph classification (Kashima & Inokuchi, 2002; Glavaš & Šnajder, 2013; Zhang et al., 2013; Jie et al., 2014) and clustering (Clarisó & Cabot, 2018; Tepeli et al., 2020). These applications involve more *local* decisions, where learning algorithms typically make inferences by examining the local structures of a graph. Certain kernels are designed to capture the structural information of graphs, such as the Weisfeiler-Leman subtree kernel (WL kernel; Shervashidze et al., 2011), which naturally lends itself to isomorphism testing (Weisfeiler & Leman, 1968). However, there exists a gap between isomorphism testing and subgraph isomorphism counting: the former is a binary problem, while the latter is a #P problem. Furthermore, isomorphism testing only considers the global structure histograms, whereas subgraph isomorphism counting requires analyzing the local structure combinations. Nonetheless, we can still utilize

graph kernels to approximate the number of isomorphic subgraphs using kernel values among thousands of graphs to represent substructures implicitly. This solution could be feasible because kernel functions and Gram matrix construction are cost-effective. With neighborhood-aware techniques and kernel tricks, we can further elevate the performance of graph kernels, making them comparable to neural networks. Code and data will be released upon publication.

## 2 Related Work

Subgraph isomorphism search, which involves finding all identical bijections, poses a more challenging problem and has been proven to be NP-complete. Numerous subgraph isomorphism algorithms have been developed, including backtracking-based algorithms and indexing-based algorithms. Ullmann's algorithm (Ullmann, 1976) is the first and the most straightforward, which enumerates all possible mappings and prunes non-matching mappings as early as possible. Several heuristic strategies have been proposed to reduce the search space and improve efficiency, such as VF2 (Cordella et al., 2004b), VF2++ (Jüttner & Madarasi, 2018), VF3 (Carletti et al., 2018), RI (Bonnici et al., 2013), QuickSI (Shang et al., 2008), TurboIso (Han et al., 2013), and BoostIso (Ren & Wang, 2015). Some algorithms are specifically designed and optimized for particular applications and database engines Graph query languages, such as GraphGrep (Giugno & Shasha, 2002) and GraphQL (He & Singh, 2008), represent patterns as hash-based fingerprints and use overlapping label-paths in the depth-first-search tree to represent branches. Various composition and selection operators can be designed based on graph structures and graph algebras. Indexing techniques play a crucial role in this area, like gIndex (Yan et al., 2004), FG-Index (Cheng et al., 2007), and CT-Index (Klein et al., 2011). Another significant direction is approximating the number of subgraph isomorphisms, striking a balance between accuracy and efficiency. Sampling techniques (Wernicke, 2005; Ribeiro & Silva, 2010; Jha et al., 2015) and color coding (Alon et al., 1995; Bressan et al., 2019) are commonly employed. However, most methods focus on homogeneous graphs and induced subgraphs, which limits their applications in real scenarios.

Graph neural networks (GNNs) can capture rich structural information of graphs, and researchers have explored their potential in subgraph matching. The message passing framework is one such technique that leverages the representation of a neighborhood as a multiset of features and aggregates neighbor messages to find functional groups in chemical molecules (Gilmer et al., 2017). Additionally, certain substructures in social networks enhance the effectiveness of recommender systems (Ying et al., 2018). Subsequently, researchers have employed graph neural networks for subgraph counting and matching purposes. Liu et al. (2020) developed a comprehensive and unified end-to-end framework that combines sequence and graph models to count subgraph isomorphisms. Ying et al. (2020) utilized graph neural networks to embed vertices and employed a voting algorithm to match subgraphs using the acquired representations. Chen et al. (2020) conducted a theoretical analysis of the upper limit of $k$-WL and similar message passing variants.

Given that the message-passing framework simulates the process of the Weisfeiler-Leman algorithm, it is still under research whether general graph kernels can be used to predict the numbers of subgraph isomorphism. One of the mainstream paradigms in the design of graph kernels is to present and compare local structures. The principal idea is to encode graphs into sparse vectors, and similar topologies should have similar representations. For example, the represented objects can be bags of components, e.g., triangles (Shervashidze et al., 2009), paths (Borgwardt & Kriegel, 2005), or neighborhood (Shervashidze et al., 2011). However, simple structures have limited the discriminative power of classifiers because two different structures may result in the same representations. Therefore, people explore higher-order structures (Shervashidze et al., 2011; Morris et al., 2020), which usually come with exponential costs, so many other graph kernels turn to generalization (Schulz et al., 2022) and efficiency (Bause & Kriege, 2022) with the loose of guidance.

## 3 Background and Motivations

### 3.1 Problem Definition

Let $\mathcal{G} = (\mathcal{V}_\mathcal{G}, \mathcal{E}_\mathcal{G}, \mathcal{X}_\mathcal{G}, \mathcal{Y}_\mathcal{G})$ be a *graph* with a vertex set $\mathcal{V}_\mathcal{G}$ and each vertex with a different *vertex id*, an edge set $\mathcal{E}_\mathcal{G} \subseteq \mathcal{V}_\mathcal{G} \times \mathcal{V}_\mathcal{G}$, a label function $\mathcal{X}_\mathcal{G}$ that maps a vertex to a set of *vertex labels*, and a label function $\mathcal{Y}_\mathcal{G}$ that maps an edge to a set of *edge labels*. To simplify the statement, we let $\mathcal{Y}_\mathcal{G}((u,v)) = \emptyset$ (where $\emptyset$

corresponds an empty set) if $(u, v) \notin \mathcal{E}_\mathcal{G}$. A *subgraph* of $\mathcal{G}$, denoted as $\mathcal{G}_S$, is any graph with $\mathcal{V}_{\mathcal{G}_S} \subseteq \mathcal{V}_\mathcal{G}$, $\mathcal{E}_{\mathcal{G}_S} \subseteq \mathcal{E}_\mathcal{G} \cap (\mathcal{V}_{\mathcal{G}_S} \times \mathcal{V}_{\mathcal{G}_S})$ satisfying $\forall v \in \mathcal{V}_{\mathcal{G}_S}, \mathcal{X}_{\mathcal{G}_S}(v) = \mathcal{X}_\mathcal{G}(v)$ and $\forall e \in \mathcal{E}_{\mathcal{G}_S}, \mathcal{Y}_{\mathcal{G}_S}(e) = \mathcal{Y}_\mathcal{G}(e)$. One of the important properties in graphs is the *isomorphism*.

**Definition 3.1 (Isomorphism)** *A graph $\mathcal{G}_1$ is isomorphic to a graph $\mathcal{G}_2$ if there is a bijection $f : \mathcal{V}_{\mathcal{G}_1} \to \mathcal{V}_{\mathcal{G}_2}$ such that:*

- $\forall v \in \mathcal{V}_{\mathcal{G}_1}, \mathcal{X}_{\mathcal{G}_1}(v) = \mathcal{X}_{\mathcal{G}_2}(f(v))$,
- $\forall v' \in \mathcal{V}_{\mathcal{G}_2}, \mathcal{X}_{\mathcal{G}_2}(v') = \mathcal{X}_{\mathcal{G}_1}(f^{-1}(v'))$,
- $\forall (u, v) \in \mathcal{E}_{\mathcal{G}_1}, \mathcal{Y}_{\mathcal{G}_1}((u, v)) = \mathcal{Y}_{\mathcal{G}_2}((f(u), f(v)))$,
- $\forall (u', v') \in \mathcal{E}_{\mathcal{G}_2}, \mathcal{Y}_{\mathcal{G}_2}((u', v')) = \mathcal{Y}_{\mathcal{G}_1}((f^{-1}(u'), f^{-1}(v')))$.

When $\mathcal{G}_1$ and $\mathcal{G}_2$ are isomorphic, we use the notation $\mathcal{G}_1 \simeq \mathcal{G}_2$ to present this and name the function $f$ as an *isomorphism*. A specific isomorphism $f$ is $\{\} \to \{\}$ when considering two empty graphs with no vertices. In addition, the *subgraph isomorphism* is more general.

**Definition 3.2 (Subgraph isomorphism)** *If a subgraph $\mathcal{G}_{1_S}$ of $\mathcal{G}_1$ is isomorphic to a graph $\mathcal{G}_2$ with a bijection $f$, we say $\mathcal{G}_1$ contains a subgraph isomorphic to $\mathcal{G}_2$ and name $f$ as a subgraph isomorphism.*

The problem of subgraph isomorphisms involves two types of subgraphs: node-induced subgraphs and edge-induced subgraphs. The former one corresponds to induced subgraph definition that requires $\mathcal{E}_{\mathcal{G}_S} = \mathcal{E}_\mathcal{G} \cap (\mathcal{V}_{\mathcal{G}_S} \times \mathcal{V}_{\mathcal{G}_S})$, while the latter one corresponds to the general definition of subgraphs. To make it easier to generalize, we assume that all subgraphs mentioned here are edge-induced.

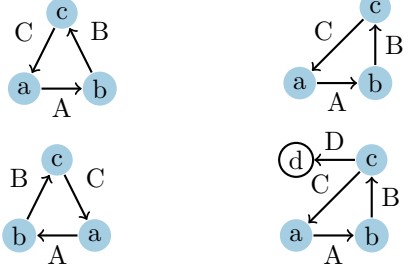

(a) Isomorphism.       (b) Subgraph isomorphism.

Figure 1: Examples of the isomorphism and subgraph isomorphism, where letters indicate labels.

### 3.2 Graph
### Isomorphism Test and Representation Power

Graph isomorphism tests are used to determine whether two graphs are isomorphic, which are useful in various fields such as computer science, chemistry, and mathematics.

**Definition 3.3 (Graph Isomorphism Test)** *A graph isomorphism test is a function $\chi : \Sigma \times \Sigma \to \{0, 1\}$ that determines whether two graphs are isomorphic, where $\Sigma$ is the graph set.*

Ideally, a perfect graph isomorphism test should be able to distinguish **all** graph pairs, i.e., $\forall \mathcal{G}_i, \mathcal{G}_j \in \Sigma : \chi(\mathcal{G}_i, \mathcal{G}_j) = 1 \Leftrightarrow \mathcal{G}_i \simeq \mathcal{G}_j$. The graph isomorphism problem is a well-known computational problem that belongs to the class of NP problems. Nonetheless, there are heuristic techniques that can solve the graph isomorphism tests for most practical cases. For example, the *Weisfeiler-Leman algorithm* (WL algorithm; Weisfeiler & Leman, 1968) is a heuristic test for graph isomorphism that assigns colors to vertices of graphs iteratively:

$$c_v^{(t+1)} = \texttt{Color}(c_v^{(t)}, \boldsymbol{N}_v^{(t)}),$$
$$\boldsymbol{N}_v^{(t)} = \{\texttt{Color}(c_u^{(t)}, \{c_{(u,v)}^{(t)}\}) | u \in \mathcal{N}_v\},$$
$$c_v^{(0)} = \mathcal{X}_\mathcal{G}(v),$$
$$c_{(u,v)}^{(t)} = \mathcal{Y}_\mathcal{G}((u, v)),$$

where $c_v^{(t)}$ is the color of vertex $v$ at the $t$-th iteration, $\mathcal{N}_v$ is $v$'s neighbor collection, $\mathcal{X}_\mathcal{G}(v)$ is the vertex label of $v$ in graph $\mathcal{G}$, $\mathcal{Y}_\mathcal{G}((u, v))$ is the edge label of $(u, v)$ in graph $\mathcal{G}$, and $\texttt{Color}$ is a function to assign colors to vertices. The time complexity of color assignment is $\mathcal{O}(|\mathcal{E}_\mathcal{G}|)$ for each iteration. Given two graphs $\mathcal{G}_i$ and $\mathcal{G}_j$, the WL algorithm refines colors to vertices of $\mathcal{G}_i$ and $\mathcal{G}_j$ in parallel and then compares the resulting colors of vertices in the two graphs.

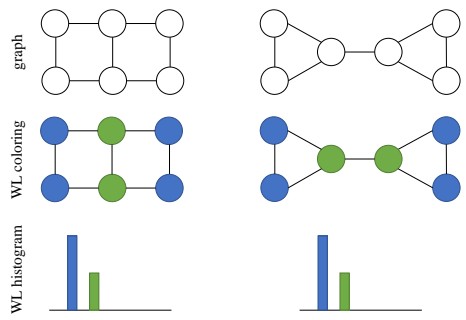 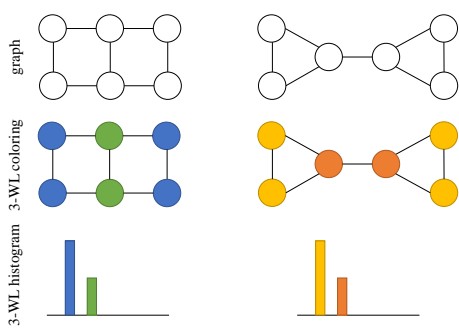

(a) Example of the non-isomorphic graphs with the same WL color histogram.

(b) Example of the non-isomorphic graphs with different 3-WL color histograms.

Figure 2: Example of the non-isomorphic graphs with the same WL color histogram but different 3-WL color histograms.

The WL algorithm is guaranteed to terminate after a finite number of iterations. It determines that two graphs are isomorphic only if the colors of vertices in both graphs are the same when the algorithm finishes, here "same" refers to having identical color histograms. However, the WL algorithm is unable to distinguish all non-isomorphic graphs, as demonstrated in Figure 2a. Therefore, it is crucial to comprehend the **representation power** of this algorithm. What makes the algorithm potent for isomorphism testing is its injective color refinement, which takes into account the neighborhood of each vertex. The neighbor collection $\mathcal{N}_v$ is defined based on the 1-hop neighborhood, meaning that color refinement relies solely on the local information of each vertex. Hence, an extension of the WL algorithm that considers higher-order interactions between vertices is expected to possess greater power for isomorphism testing, which are called *k-dimensional Weisfeiler-Leman algorithm* (Cai et al., 1992). It iteratively assigns colors to the *k*-tuples ($k \geq 2$) of $\mathcal{V}_{\mathcal{G}}^k$ as follows:

$$c_{\boldsymbol{v}}^{(t+1)} = \texttt{Color}(c_{\boldsymbol{v}}^{(t)}, \boldsymbol{N}_{\boldsymbol{v}}^{(t)}),$$

$$\boldsymbol{N}_{\boldsymbol{v}}^{(t)} = \bigcup_{j=1}^{k} \{\texttt{Color}(c_{\boldsymbol{u}}^{(t)}, \{c_{(\boldsymbol{u},\boldsymbol{v})}^{(t)}\}) | \boldsymbol{u} \in \mathcal{N}_{\boldsymbol{v}}^j\},$$

$$c_{\boldsymbol{v}}^{(0)} = \texttt{Color}(\mathcal{X}_{\mathcal{G}}(\boldsymbol{v}[k]), \{\mathcal{X}_{\mathcal{G}}(\boldsymbol{v}) | w \in \boldsymbol{v} - \{\boldsymbol{v}[k]\}\}),$$

$$c_{(\boldsymbol{u},\boldsymbol{v})}^{(t)} = \begin{cases} \mathcal{Y}_{\mathcal{G}}((u,v)) & \text{if } u = \boldsymbol{v} - \boldsymbol{u} \text{ and } v = \boldsymbol{u} - \boldsymbol{v}, \\ \emptyset & \text{otherwise.} \end{cases},$$

where $\boldsymbol{v}$ refers to a tuple of $k$ vertices in $\mathcal{V}_{\mathcal{G}}$, $\boldsymbol{v}[j]$ denotes the $j$-th element of the tuple $\boldsymbol{v}$, $\mathcal{N}_{\boldsymbol{v}}^j$ ($1 \leq j \leq k$) is the neighbor collection of $\boldsymbol{v}$ in which only the $j$-th elements are different (i.e., $\mathcal{N}_{\boldsymbol{v}}^j = \{\boldsymbol{u} | (\forall i \neq j \ \boldsymbol{u}[i] = \boldsymbol{v}[i]) \wedge (\boldsymbol{u}[j] = \boldsymbol{v}[j])\}$), $\mathcal{X}_{\mathcal{G}}(\boldsymbol{v}[k])$ is the vertex label of $\boldsymbol{v}[k]$ in graph $\mathcal{G}$, $\mathcal{X}_{\mathcal{G}}(\boldsymbol{v})$ is the vertex label of $\boldsymbol{v}$ in graph $\mathcal{G}$, $\mathcal{Y}_{\mathcal{G}}((u,v))$ is the edge label of $(u,v)$ in graph $\mathcal{G}$, $\boldsymbol{v} - \boldsymbol{u}$ indicates the difference of two tuples, and $\texttt{Color}$ is a function to assign colors to tuples. The $k$-WL algorithm is also guaranteed to terminate after a finite number of iterations. Figure 2b demonstrates the same example in Figure 2a with different 3-WL color histograms, despite having the same WL color histogram. This suggests that the 3-WL algorithm can distinguish more non-isomorphic graphs than the WL algorithm, which is expected to obtain strictly stronger representation power. It is worth noting that the complexity of the $k$-WL algorithm increases exponentially because it constructs $|\mathcal{V}_{\mathcal{G}}|^k$ tuples and at most $|\mathcal{V}_{\mathcal{G}}|^{2k}$ connections, which are regarded as "vertices" and "edges" in the $k$-tuple graph. Thus, the complexity of color assignment becomes $\mathcal{O}(|\mathcal{V}_{\mathcal{G}}|^{2k})$.

### 3.3 Graph Kernels

However, there is a gap between the power of representation and subgraph isomorphisms. While the power aims to distinguish non-isomorphic graph pairs, counting subgraph isomorphisms presents a greater challenge, which is a combinatorial problem depending on substructures. Therefore, our objective is to approximate subgraph isomorphism counting through representation learning and optimization as regression. The iso-

morphism test can be seen as a hard indicator function that determines whether an isomorphism exists, which can be extended as a kernel function for subgraph isomprhism counting. A *kernel function* is designed to measure the similarity between two objects.

**Definition 3.4 (Kernel)** *A function $k : \Sigma \times \Sigma \to \mathbb{R}$ is called a kernel over an non-empty set $\Sigma$.*

An even more crucial concept is the selection of a kernel function $k$ in a manner that allows for the existence of a *feature map $h$* from the set $\Sigma$ to a Hilbert space $\mathbb{H}$ equipped with an inner product. This feature map should satisfy $\forall \mathcal{G}_i, \mathcal{G}_j \in \Sigma : k(\mathcal{G}_i, \mathcal{G}_j) = \langle h(\mathcal{G}_i), h(\mathcal{G}_j) \rangle$. This space $\mathbb{H}$ is referred to as the *feature space*. A group of kernel functions known as *graph kernels* (GKs) are employed to compute the similarity between two graphs by taking them as input.

Neural networks have been regarded as effective feature extractors and predictors, and sequence and graph neural networks can be aligned with the kernel functions (Lei et al., 2017; Xu et al., 2019). Liu et al. (2020) proposed a unified end-to-end framework for sequence models and graph models to directly predict the number of subgraph isomorphisms rather than the similarities, further illustrating the practical success.

## 4 Beyond the Representation Power Limitation via Implicit Correlations

Constructing neural networks to directly predict the number of subgraph isomorphisms has been shown effective and efficient (Liu et al., 2020; Yu et al., 2023). But transforming $\Sigma$ to a limited-dimensional space $\mathbb{H}$ remains challenging in optimization, and it has been shown bounded in theory and practice (Chen et al., 2020; Liu & Song, 2022). Therefore, we turn to other directions to leverage implicit structure correlations to make predictions.

### 4.1 Gram Matrix Construction

Given a set of graphs $\mathcal{G}_1, \mathcal{G}_2, \cdots, \mathcal{G}_D \in \Sigma$, the *kernel matrix $\boldsymbol{K}$* is defined as:

$$\boldsymbol{K} = \begin{bmatrix} k(\mathcal{G}_1, \mathcal{G}_1) & k(\mathcal{G}_1, \mathcal{G}_2) & \cdots & k(\mathcal{G}_1, \mathcal{G}_D) \\ k(\mathcal{G}_2, \mathcal{G}_1) & k(\mathcal{G}_2, \mathcal{G}_2) & \cdots & k(\mathcal{G}_2, \mathcal{G}_D) \\ \vdots & \vdots & \ddots & \vdots \\ k(\mathcal{G}_D, \mathcal{G}_1) & k(\mathcal{G}_D, \mathcal{G}_2) & \cdots & k(\mathcal{G}_D, \mathcal{G}_D) \end{bmatrix}, \tag{1}$$

$$s.t. \boldsymbol{K}_{ij} = k(\mathcal{G}_i, \mathcal{G}_j) = \langle h(\mathcal{G}_i), h(\mathcal{G}_j) \rangle. \tag{2}$$

The kernel matrix $\boldsymbol{K} \in \mathbb{R}^{D \times D}$ is also called the *Gram matrix*. Different kernel functions emphasize specific structural properties of graphs. For instance, *Shortest-path Kernel* (SP) (Borgwardt & Kriegel, 2005) decomposes graphs into shortest paths and compares graphs according to their shortest paths, such as path lengths and endpoint labels. Instead, Graphlet Kernels (Shervashidze et al., 2009) compute the distribution of small subgraphs (i.e., wedges and triangles) under the assumption that graphs with similar graphlet distributions are highly likely to be similar. The *Weisfeiler-Leman subtree kernel* (WL kernel) belongs to a family of graph kernels denoted as ($k$-WL) (Shervashidze et al., 2011), where $k$ indicates the element size during the color assignment.

Let's take the WL kernel as an example. It is a popular graph kernel based on the WL algorithm mentioned in § 3.2 upon 1-hop neighborhood aggregation to assign finite integer labels $\mathbb{S}$, i.e., $h_{\mathrm{KL}}$: $\mathcal{G} \to \left\| \Large\|_{s \in \mathbb{S}} \#\{v \in \mathcal{V}_{\mathcal{G}} | c_v = s\}$ such that Color: $v \in \mathcal{V}_{\mathcal{G}} \to c_v \in \mathbb{S}$. Usually, the convergence of the colors for different graphs occurs in different iterations, so it is hard to determine a specific "finite" number of iterations. Thus, the WL kernel is often manually set to a particular number $T$:

$$k_{\mathrm{WL}}(\mathcal{G}_i, \mathcal{G}_j) = \langle h_{\mathrm{WL}}(\mathcal{G}_i), h_{\mathrm{WL}}(\mathcal{G}_j) \rangle = \left\langle \Big\|_{t=0}^{T} \boldsymbol{C}_{\mathcal{V}_{\mathcal{G}_i}}^{(t)}, \Big\|_{t=0}^{T} \boldsymbol{C}_{\mathcal{V}_{\mathcal{G}_j}}^{(t)} \right\rangle, \tag{3}$$

where $\boldsymbol{C}_{\mathcal{V}_{\mathcal{G}_i}}^{(t)}$ is the color histogram (a color vector that counts the number of occurrences of vertex colors)

of $\mathcal{G}_i$ at iteration $t$, i.e., $\boldsymbol{C}_{\mathcal{V}_{\mathcal{G}_i}}^{(t)} = \Big\|_{s \in \mathbb{S}} \#\{v \in \mathcal{V}_{\mathcal{G}_i} | \boldsymbol{c}_v^{(t)} = s\}$. Note that there is no overlap between the colors at different iterations such that color vectors $\{\boldsymbol{C}_{\mathcal{V}_{\mathcal{G}_i}}^{(t)} | 0 \leq t \leq T\}$ are orthogonal to each other. Hence, the kernel is efficient in computing by reducing a sum of inner products:

$$k_{\text{WL}}(\mathcal{G}_i, \mathcal{G}_j) = \sum_{t=0}^{T} \boldsymbol{C}_{\mathcal{V}_{\mathcal{G}_i}}^{(t)}{}^{\top} \boldsymbol{C}_{\mathcal{V}_{\mathcal{G}_j}}^{(t)}. \tag{4}$$

It is worth noting that the Gram matrix does not explicitly maintain the graph structure and substructure information. But this information can be implicitly captured within the matrix.

The running time for a single color vector is $\mathcal{O}(T \cdot |\mathcal{E}_{\mathcal{G}_i}|)$, and the running time for the dot product is $\mathcal{O}(T \cdot (|\mathcal{V}_{\mathcal{G}_i}| + |\mathcal{V}_{\mathcal{G}_j}|))$. Therefore, the running time for the WL kernel and the Gram matrix is $\mathcal{O}(T \cdot M \cdot D + T \cdot N \cdot D^2)$, where $N$ and $M$ represent the maximum number of vertices and edges among the $D$ graphs, respectively. For a $k$-WL kernel, the time complexity is $\mathcal{O}(T \cdot N^{2k} \cdot D + T \cdot N \cdot D^2)$.

### 4.2 Neighborhood Information in the Hilbert Space

Since the graph kernel is a function of the graph representation, the graph structure is expected to be preserved in the Hilbert space. However, the hash function in the WL kernel family does not capture the neighbors of a node. For example, $c_u^{(t)}$ and $c_v^{(t)}$ would be different if $u \in \mathcal{V}_{\mathcal{G}i}$ and $v \in \mathcal{V}_{\mathcal{G}_j}$ have different neighbors (more precisely, at least one neighbor is different). Nevertheless, the subset of neighbors is essential for examining isomorphisms, as the inclusion relation is a necessary condition for subgraph isomorphisms. We modify the color assignment algorithm in the WL kernel family to incorporate neighborhood information in the Hilbert space or record it in the graph histogram. The modified WL kernel is called *neighborhood-information-extraction WL kernel* (NIE-WL kernel). The neighborhood-aware color assignment algorithm is described in Algorithm 1. The only change is the addition of pairwise colors to the color histogram. This pairwise color depends on the edges and the latest node colors, without affecting the original color assignment. As a result, the color histogram becomes more expressive, as it can record neighborhood information.

It is clear that the NIE-WL kernel should have the same expressive power as the WL kernel, but the histogram of the NIE-WL kernel records $|\mathcal{V}_{\mathcal{G}}| + |\mathcal{E}_{\mathcal{G}}|$ colors instead of $|\mathcal{V}_{\mathcal{G}}|$ colors. The additional $|\mathcal{E}_{\mathcal{G}}|$ colors (denoted as $\boldsymbol{C}_{\mathcal{E}_{\mathcal{G}_i}}^{(t)}$ for the $t$-th iteration) can be used in constructing the Gram matrix, where neighborhood information is preserved. If we decompose the NIE-WL kernel into the WL kernel and the neighborhood-information-extraction kernel (denoted as NIE), we can get:

$$k_{\text{NIE-WL}}(\mathcal{G}_i, \mathcal{G}_j) = \Big\langle \Big\|_{t=0}^{T} \boldsymbol{C}_{\mathcal{V}_{\mathcal{G}_i}}^{(t)} \| \boldsymbol{C}_{\mathcal{E}_{\mathcal{G}_i}}^{(t)}, \Big\|_{t=0}^{T} \boldsymbol{C}_{\mathcal{V}_{\mathcal{G}_j}}^{(t)} \| \boldsymbol{C}_{\mathcal{E}_{\mathcal{G}_j}}^{(t)} \Big\rangle = \sum_{t=0}^{T} (\boldsymbol{C}_{\mathcal{V}_{\mathcal{G}_i}}^{(t)} \| \boldsymbol{C}_{\mathcal{E}_{\mathcal{G}_i}}^{(t)})^{\top} (\boldsymbol{C}_{\mathcal{V}_{\mathcal{G}_j}}^{(t)} \| \boldsymbol{C}_{\mathcal{E}_{\mathcal{G}_j}}^{(t)})$$

$$= \sum_{t=0}^{T} \boldsymbol{C}_{\mathcal{V}_{\mathcal{G}_i}}^{(t)}{}^{\top} \boldsymbol{C}_{\mathcal{V}_{\mathcal{G}_j}}^{(t)} + \sum_{t=0}^{T} \boldsymbol{C}_{\mathcal{E}_{\mathcal{G}_i}}^{(t)}{}^{\top} \boldsymbol{C}_{\mathcal{E}_{\mathcal{G}_j}}^{(t)} = k_{\text{WL}}(\mathcal{G}_i, \mathcal{G}_j) + k_{\text{NIE}}(\mathcal{G}_i, \mathcal{G}_j). \tag{5}$$

In other words, the NIE-WL kernel is the linear combination of the WL kernel and the neighborhood-information-extraction kernel. This also implies that the NIE-WL kernel is a hybrid kernel function, incorporating information beyond feature transformations.

### 4.3 Kernel Tricks

Graph kernels are typically characterized as *positive semi-definite kernels*. Consequently, these kernels $\boldsymbol{K}$ possess the *reproducing property*:

$$\theta(\mathcal{G}_i) = \sum_{j=1}^{D} \boldsymbol{K}_{ij} \theta(\mathcal{G}_j) = \langle \boldsymbol{K}_i, \boldsymbol{\theta} \rangle, \tag{6}$$

where $\theta$ is a function belonging to a new feature space $\mathbb{H}'$, and $\boldsymbol{\theta} = [\theta(\mathcal{G}_1), \theta(\mathcal{G}_2), \cdots, \theta(\mathcal{G}_D)]$ is the vectorized

---

**Algorithm 1** Neighborhood-aware color assignment algorithm.

---

**input** a directed graph $\mathcal{G} = (\mathcal{V}_\mathcal{G}, \mathcal{E}_\mathcal{G}, \mathcal{X}_\mathcal{G}, \mathcal{Y}_\mathcal{G})$, a fixed integer $T$
1: initialize the color of each node $v$ in $\mathcal{V}_\mathcal{G}$ as $c_v^{(0)} = \mathcal{X}_\mathcal{G}(v)$ and color of each edge $e$ in $\mathcal{E}_\mathcal{G}$ as $c_e = \mathcal{Y}_\mathcal{G}(e)$
2: **for** iter $t$ from 1 to $T$ **do**
3:     create a new color counter $\mathcal{C}^{(t)}$ and initialize $\mathcal{C}^{(t)} = \emptyset$
4:     **for** each $v$ in $\mathcal{V}_\mathcal{G}$ **do**
5:       create a color multi-set $\boldsymbol{N}_v^{(t)}$ and initialize $\boldsymbol{N}_v^{(t)} = \emptyset$
6:       **for** each neighbor $u$ in v's neighbor set $\mathcal{N}_v$ **do**
7:         add the neighbor color $\texttt{Color}(c_u^{(t)}, \{c_{u,v}\})$ to $\boldsymbol{N}_v^{(t)}$
8:         calculate the pair-wise color $\texttt{Color}(c_v^{(t)}, \{\texttt{Color}(c_u^{(t)}, \{c_{u,v}\})\})$ and record it in the color counter $\mathcal{C}^{(t)}$
           //{neighborhood information recording}
9:       **end for**
10:       calculate the set-wise color $\texttt{Color}(c_v^{(t)}, \boldsymbol{N}_v^{(t)})$, record it in the color counter $\mathcal{C}^{(t)}$, and update $c_v^{(t+1)}$     //{original color assignment}
11:     **end for**
12: **end for**
**output** the graph color histogram $\bigcup_{t=0}^{T} \{\mathcal{C}^{(t)}\}$

---

representation of $\theta$. The space $\mathbb{H}'$ is known as the *reproducing kernel Hilbert space* (RKHS) and does not require explicit construction.

Based on the definition of kernels and the reproducing property, a graph $\mathcal{G}_i$ can be represented as $\boldsymbol{g}_i$ in the Hilbert space (more precisely, RKHS) according to Eq. (6). We then embed the resulting graph representation into another Hilbert space, $\mathbb{H}'$, using another kernel function $k'$. We consider the following two popular kernel functions.

**Polynomial Kernel** The *polynomial kernel* is defined as $k_{\text{Poly}}(\mathcal{G}_i, \mathcal{G}_j) = (\boldsymbol{g}_i^\top \boldsymbol{g}_j + 1)^p$, where $p \in \mathbb{N}$ is a positive integer. In practice, explicitly computing the polynomial kernel matrix such that $\boldsymbol{K}_{i,j} = k_{\text{Poly}}(\boldsymbol{g}_i, \boldsymbol{g}_j)$ is infeasible due to the high dimensionality of the Hilbert space. Instead, we employ the kernel trick to compute the polynomial kernel matrix implicitly:

$$k_{\text{Poly}}(\mathcal{G}_i, \mathcal{G}_j) = (\boldsymbol{g}_i^\top \boldsymbol{g}_j + 1)^p = \sum_{k=0}^{p} \frac{p!}{k!(p-k)!} (\boldsymbol{g}_i^\top \boldsymbol{g}_j)^{p-k} 1^k = (\boldsymbol{K}_{i,j} + 1)^p. \tag{7}$$

**Gaussian Kernel** A polynomial kernel transforms the graph representation into a higher dimensional space $\binom{|\Sigma|+p}{p}$. However, the polynomial kernel is sensitive to the parameter $p$, which may result in an overflow issue when $p$ is too large. A popular kernel function that maps the graph representation into an infinite-dimensional space is named *Gaussian kernel* or *radial basis function kernel*, i.e., $k_{\text{RBF}}(\mathcal{G}_i, \mathcal{G}_j) = \exp(-\frac{\|\boldsymbol{g}_i - \boldsymbol{g}_j\|^2}{2\sigma^2})$, where $\sigma$ is a positive real number. We also have a trick to compute the Gaussian kernel matrix implicitly:

$$k_{\text{RBF}}(\mathcal{G}_i, \mathcal{G}_j) = \exp(-\frac{\|\boldsymbol{g}_i - \boldsymbol{g}_j\|^2}{2\sigma^2}) = \exp(-\frac{\boldsymbol{g}_i^\top \boldsymbol{g}_i - 2\boldsymbol{g}_i^\top \boldsymbol{g}_j + \boldsymbol{g}_j^\top \boldsymbol{g}_j}{2\sigma^2}) = \exp(-\frac{\boldsymbol{K}_{i,i} - 2\boldsymbol{K}_{i,j} + \boldsymbol{K}_{j,j}}{2\sigma^2}). \tag{8}$$

By employing the aforementioned implicit computation tricks, the kernel transformations become efficient and scalable through matrix operations. Hybrid kernel functions can be obtained by combining different graph kernels. For instance, a hybrid kernel function with the WL kernel and RBF kernel is Eq. (9), where $\boldsymbol{K}_{\text{WL}}$ is the Gram matrix with respect to the WL kernel.

$$k_{\text{WL,RBF}}(\mathcal{G}_i, \mathcal{G}_j) = \exp(-\frac{\boldsymbol{K}_{\text{WL}i,i} - 2\boldsymbol{K}_{\text{WL}i,j} + \boldsymbol{K}_{\text{WL}j,j}}{2\sigma^2}). \tag{9}$$

# 5 Experiment

## 5.1 Experimental Setup

**Evaluation**  We regard subgraph isomorphism counting with graph kernels as a machine learning problem. Since we model subgraph isomorphism counting as a regression problem with Gram matrices, we use the SVM (Chang & Lin, 2011) and Ridge (Hoerl & Kennard, 1970) implemented in the scikit-learn package. We assess models based on the root mean squared error (RMSE) and the mean absolute error (MAE). We collect the most popular datasets for graph classification, as graph properties are often determined by substructures within the graph. In order to obtain meaningful and challenging predictions, we enumerate all vertex label permutations and edge permutations from the 3-stars, triangles, tailed triangles, and chordal cycles. Furthermore, to enhance the quality of the data, we filter out patterns with an average frequency of less than 1.0 across the entire dataset. Detailed settings can be found in Appendix A. Our approach to graph kernels involves substructures from different levels:

- **Paths**: Shortest-path Kernel (Borgwardt & Kriegel, 2005) decomposes graphs into shortest paths and compares these paths.

- **Wedges and triangles**: Graphlet Kernels (Shervashidze et al., 2009) compute the distribution of graphlets of size 3, which consist of wedges and triangles.

- **Whole graphs**: Weisfeiler-Leman Optimal Assignment Kernel (WLOA) (Kriege et al., 2016) improves the WL kernel by capitalizing on the theory of valid assignment kernels.

- **High-order neighborhood**: $k$-WL Kernels (Shervashidze et al., 2011) measure the histogram of $k$-combinations by assigning colors to $k$-tuples, while $\delta$-$k$ WL kernels record

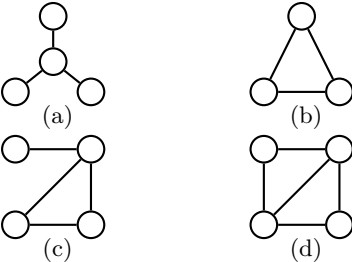

Figure 3: Patterns considered in experiments: (a) 3-star, (b) triangle, (c) tailed triangle, and (d) chordal cycle.

the number of $k$-tuples with the same color. $\delta$-$k$-LWL and $\delta$-$k$-LWL$^+$ focus on local structures of the graph instead of the whole graph (Morris et al., 2020).

We also apply polynomial and Gaussian kernel transformations to the kernels mentioned above. Besides, we incorporate the neighborhood-aware color assignment to $k$-WL Kernels and their variants. We provide two trivial baselines and neural baselines as references (please refer to Appendix B.1).

**Efficient Implementation**  Graph kernels are implemented in C++ with C++11 features and the -O2 flag. In addition to the technical aspects, we also focus on training efficiency. The simplest input for regressors is the original graph kernel matrix of size $D \times D$. However, this kernel matrix only contains graphs in the entire dataset, without any information about patterns. It is necessary to incorporate pattern structure information during training. Assuming we have $Q$ patterns, we need to repeatedly construct $Q$ kernel matrices of size $(1+D) \times (1+D)$. In fact, the $D \times D$ submatrix is the same for all $Q$ kernel matrices because it is irrelevant to the patterns. Therefore, it is recommended that we construct a matrix of size $(Q+D) \times (Q+D)$ only once and then repeatedly slice the submatrix to obtain information about the $D$ graphs and the specific pattern for prediction, as demonstrated in Figure 4.

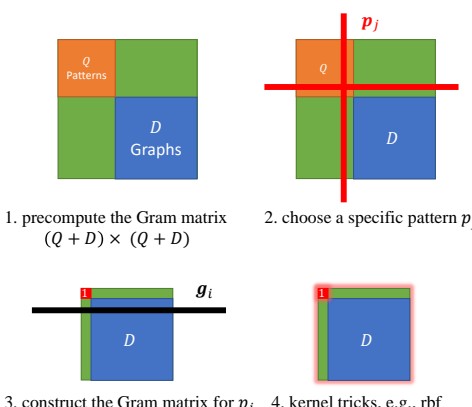

1. precompute the Gram matrix $(Q+D) \times (Q+D)$

2. choose a specific pattern $p_j$

3. construct the Gram matrix for $p_j$ $(1+D) \times (1+D)$

4. kernel tricks, e.g., rbf

Figure 4: Efficient implementation.

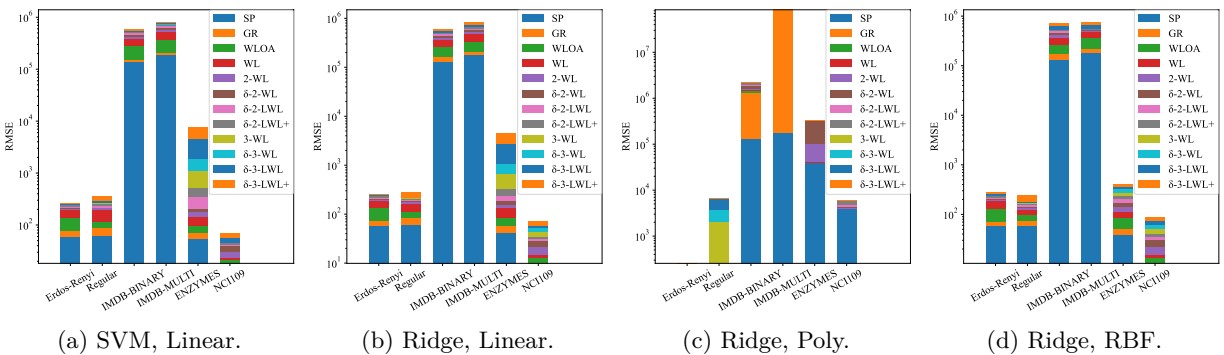

| (a) SVM, Linear. | (b) Ridge, Linear. | (c) Ridge, Poly. | (d) Ridge, RBF. |

Figure 5: Illustration on subgraph isomorphism counting, where the out-of-memory ("OOM") is regarded as zero in the plots.

Table 1: Performance on subgraph isomorphism counting, where $k$-WL‡ represents the best model in the kernel family, and the best and second best are marked in red and blue, respectively.

| | Homogeneous | | | | | | | | Heterogeneous | | | |
|---|---|---|---|---|---|---|---|---|---|---|---|---|
| Models | *Erdős-Renyi* | | *Regular* | | IMDB-BINARY | | IMDB-MULTI | | *ENZYMES* | | *NCI109* | |
| | RMSE | MAE | RMSE | MAE | RMSE | MAE | RMSE | MAE | RMSE | MAE | RMSE | MAE |
| Zero | 92.532 | 51.655 | 198.218 | 121.647 | 138262.003 | 37041.171 | 185665.874 | 33063.770 | 64.658 | 25.110 | 16.882 | 7.703 |
| Avg | 121.388 | 131.007 | 156.515 | 127.211 | 133228.554 | 54178.671 | 182717.385 | 53398.301 | 59.589 | 31.577 | 14.997 | 8.622 |
| TXL | 10.861 | 7.105 | 15.263 | 10.721 | 15369.186 | 3170.290 | 19706.248 | 3737.862 | 25.912 | 11.284 | 5.482 | 2.823 |
| RGCN | 9.386 | 5.829 | 14.789 | 9.772 | 46074.355 | 13498.414 | 69485.242 | 12137.598 | 23.715 | 11.407 | **1.217** | 0.622 |
| RGIN | 6.063 | 3.712 | 13.554 | 8.580 | 31058.764 | 6445.103 | 26546.882 | 4508.339 | **8.119** | **3.783** | **0.511** | **0.292** |
| CompGCN | 6.706 | 4.274 | 14.174 | 9.685 | 32145.713 | 8576.071 | 26523.880 | 7745.702 | 14.985 | 6.438 | 1.271 | 0.587 |
| | | | | | Ridge, Linear | | | | | | | |
| SP | 58.721 | 34.606 | 60.375 | 41.110 | 131672.705 | 56058.593 | 181794.702 | 54604.564 | 43.007 | 14.422 | 4.824 | 2.268 |
| GR | 14.067 | 7.220 | 23.775 | 12.172 | 30527.764 | 7894.568 | 30980.135 | 6054.027 | 14.557 | 5.595 | 5.066 | 2.066 |
| WLOA | 58.719 | 34.635 | 25.906 | 17.003 | 96887.226 | 28849.659 | 117828.698 | 25808.362 | 28.911 | 11.807 | 3.142 | 1.142 |
| WL | 58.719 | 34.635 | 56.045 | 33.891 | 107500.276 | 41523.359 | 147822.358 | 49244.753 | 46.466 | 14.920 | 1.896 | 0.746 |
| 2-WL‡ | 9.921 | 4.164 | 8.751 | 5.663 | 33336.019 | 9161.265 | 47075.541 | 13751.520 | 26.903 | 10.079 | 2.584 | 1.068 |
| 3-WL‡ | **4.096** | **1.833** | **3.975** | **2.277** | 39237.071 | 7240.730 | 76218.966 | 9022.754 | 335.940 | 13.790 | 3.872 | 1.375 |
| | | | | | Ridge, Linear, NIE | | | | | | | |
| WLOA | 58.719 | 34.635 | 25.905 | 17.003 | 33625.086 | 6009.372 | 20858.288 | 2822.391 | 23.478 | 10.037 | 3.203 | 1.133 |
| WL | 58.719 | 34.635 | 56.045 | 33.891 | 66414.032 | 17502.328 | 70013.402 | 13266.318 | 20.971 | 8.672 | 1.772 | 0.704 |
| 2-WL‡ | 9.921 | 4.164 | 8.751 | 5.663 | 14914.025 | 3671.681 | 37676.903 | 9930.398 | 97.024 | 7.191 | 1.259 | **0.539** |
| 3-WL‡ | **4.096** | **1.833** | **3.975** | **2.277** | **1808.841** | **264.480** | **1346.608** | **123.394** | 380.480 | 19.073 | OOM | OOM |
| | | | | | Ridge, RBF | | | | | | | |
| SP | 58.721 | 34.606 | 60.375 | 41.110 | 131672.705 | 56058.593 | 181794.702 | 54604.564 | 38.945 | 14.712 | 5.474 | 2.224 |
| GR | 11.670 | 5.663 | 12.488 | 5.012 | 42387.021 | 5110.985 | 41171.761 | 4831.495 | **12.883** | **5.073** | 4.804 | 1.944 |
| WLOA | 58.719 | 34.635 | 25.906 | 17.003 | 92733.105 | 28242.033 | 137300.092 | 34067.513 | 32.827 | 12.230 | 3.215 | 1.261 |
| WL | 58.719 | 34.635 | 25.905 | 17.003 | 109418.159 | 32350.523 | 112515.690 | 25035.268 | 26.313 | 10.933 | 2.227 | 0.837 |
| 2-WL‡ | 10.500 | 4.630 | 8.495 | 5.634 | 40412.745 | 5351.789 | 21910.109 | 2982.532 | 29.560 | 11.878 | 5.001 | 1.799 |
| 3-WL‡ | **4.896** | **2.536** | **4.567** | **2.745** | 89532.736 | 21918.757 | 91445.323 | 17703.656 | 43.908 | 18.509 | 10.925 | 5.320 |
| | | | | | Ridge, RBF, NIE | | | | | | | |
| WLOA | 58.719 | 34.635 | 25.906 | 17.002 | 31409.659 | 6644.798 | 19456.664 | 3892.678 | 24.429 | 10.354 | 3.163 | 1.189 |
| WL | 58.719 | 34.635 | 25.905 | 17.003 | 48568.177 | 17533.158 | 71434.770 | 20472.124 | 23.155 | 9.302 | 2.026 | 0.805 |
| 2-WL‡ | 10.500 | 4.630 | 8.495 | 5.634 | 15241.302 | 3289.949 | 30093.401 | 6593.717 | 33.838 | 13.947 | 6.619 | 2.807 |
| 3-WL‡ | **4.896** | **2.536** | **4.567** | **2.745** | **757.736** | **148.417** | **833.037** | **75.286** | 43.918 | 18.491 | OOM | OOM |

## 5.2 Empirical Results

### 5.2.1 SVM vs. Ridge

We begin by comparing the performance of SVM and Ridge (precisely, Kernel Ridge) regression without kernel tricks on subgraph isomorphism counting. This is a common practice to evaluate the performance of these two regressors. As shown in Figure 5 and Appendix B.2, the performance of the two regressors is comparable, with Ridge performing slightly better. There are two main reasons. First, Ridge is solved by Cholesky factorization in the closed form, which typically achieves better convergence than the iterative optimization of SVM. Second, the objective of Ridge is to minimize the sum of squared errors, which is more straightforward and suitable for the regression task than the $\epsilon$-insensitive loss function of SVM. Therefore, we mainly report the results of Ridge.

### 5.2.2 Kernel Tricks for Implicit Transform

In addition, we also observe an increase in errors with the polynomial kernel trick from Figure 5. The number of matched subgraphs is typically small but can be very large for specific structures, such as complete graphs or cliques. The polynomial kernel trick can easily lead to fluctuations due to extreme values, resulting in performance fluctuations and even overflow.

### 5.2.3 Effectiveness of Neighborhood Information Extraction

Explicit neighborhood information extraction (NIE) is a crucial component for handling homogeneous data by providing edge colors. However, this method is not as beneficial when applied to synthetic *Erdős-Renyi* and *Regular* datasets because the uniform distribution of neighborhoods results in uniform distributions of edge colors. As demonstrated in Table 1, incorporating NIE consistently enhances the performance of both linear and RBF kernels.

Overall, the RBF kernel combined with NIE proves to be more effective for homogeneous data, while the linear kernel is substantially improved when applied to heterogeneous data. The most significant enhancements are observed on the highly challenging *IMDB-BINARY* and *IMDB-MULTI* datasets, where the RMSE is dramatically reduced from 30,527.764 to 757.736 and from 21,910.109 to 833.037, respectively. When compared to naive baselines that predict either zeros or the training sets' expectations, the RMSE is diminished to a mere 0.5%. In addition, some kernel methods, such as GR and the 2-WL family, can provide the same or even more reliable predictions as neural methods. Moreover, NIE attains state-of-the-art performance on homogeneous data, with a relative gain of 72.1% compared with the best neural networks. As for the remaining two heterogeneous datasets, neighborhood information is still not comparable to the GR kernel in *ENZYMES*. This observation is aligned with the performance of CompGCN (Vashishth et al., 2020), where such the node-edge composition may hurt the structure presentation. RGIN (Liu et al., 2020) significantly outperforms graph kernels, indicating the future direction of advanced subset representations. These significant findings serve as a foundation for further research and advancements in the field of graph kernels, as well as other representation learning methods like graph neural networks.

### 5.3 Discussion

Experimental results demonstrate that kernel tricks can enhance the performance of Ridge regression. Additionally, NIE could relief overfitting by additional pair-wise histograms and obtain significant improvement in challenging datasets. Upon analyzing the differences between the GR kernel and our proposed NIE, high-order topologies such as triangles and wedges could be more powerful than the first-order topologies. But we still see that the 3-WL-family kernels may perform the worst on heterogeneous data. These findings can serve as a foundation for further research and advancements in the field of graph kernels and other representation learning methods such as graph neural networks.

## 6 Conclusion

We are the first to concentrate on the representation of patterns and subgraphs by utilizing a variety of graph kernels to tackle the challenge of subgraph isomorphism counting. While most graph kernels are designed for substructures, their application in approximating subgraph isomorphism counting is not straightforward. Instead, we propose constructing the Gram matrix to leverage implicit correlations. Experimental results demonstrate the effectiveness of graph kernels and kernel tricks. Additionally, neighborhood information extraction (NIE) could relieve overfitting by additional pair-wise histograms and obtain significant improvement in challenging datasets.

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

## A   Experiment Settings

### A.1   Benchmarking Datasets

We collect the most popular datasets (listed in Table 2) for graph classification, as graph properties are often determined by substructures within the graph, such as motifs. Among the datasets, *IMDB-BINARY* and *IMDB-MULTI* consist of ego-networks derived from actor collaborations in the IMDB dataset. *ENZYMES* represents macromolecules and their interactions in the bioinformatics field, which were collected from the BRENDA database Schomburg et al. (2004). *NCI109* consists of molecular compounds for interaction prediction. Additionally, we selected two synthetic homogeneous datasets, *Erdős-Renyi* and *Regular* Chen et al. (2020), for benchmarking purposes.

Table 2: Dataset statistics on subgraph isomorphism experiments. $\mathcal{P}$ and $\mathcal{G}$ corresponds to patterns and graphs.

| | | Erdős-Rényi | | Regular | | IMDB-BINARY | | IMDB-MULTI | | ENZYMES | | NCI109 |
|---|---|---|---|---|---|---|---|---|---|---|---|---|
| # Train | | 6,000 | | 6,000 | | 1,332 | | 2,000 | | 8,400 | | 6,875 |
| # Valid | | 4,000 | | 4,000 | | 1,336 | | 2,000 | | 8,400 | | 6,880 |
| # Test | | 10,000 | | 10,000 | | 1,332 | | 2,000 | | 8,400 | | 6,880 |
| Avg. Subgraph Isomorphisms | | 51.655 | | 121.647 | | 37041.171 | | 33063.770 | | 25.110 | | 7.703 |
| | Max | Avg. | Max | Avg. | Max | Avg. | Max | Avg. | Max | Avg. | Max | Avg. |
| $|\mathcal{V}_\mathcal{P}|$ | 4 | 3.8±0.4 | 4 | 3.8±0.4 | 4 | 3.75±0.4 | 4 | 3.75±0.4 | 4 | 3.9±0.3 | 4 | 4±0 |
| $|\mathcal{E}_\mathcal{P}|$ | 10 | 7.5±1.7 | 10 | 7.5±1.7 | 10 | 7.5±1.7 | 10 | 7.5±1.7 | 10 | 7.6±1.5 | 6 | 6±0 |
| $|\mathcal{X}_\mathcal{P}|$ | 1 | 1±0 | 1 | 1±0 | 1 | 1±0 | 1 | 1±0 | 3 | 2.1±0.5 | 3 | 3±0 |
| $|\mathcal{Y}_\mathcal{P}|$ | 1 | 1±0 | 1 | 1±0 | 1 | 1±0 | 1 | 1±0 | 1 | 1±0 | 1 | 1±0 |
| $|\mathcal{V}_\mathcal{G}|$ | 10 | 10±0 | 30 | 18.8±7.4 | 136 | 19.8±10.1 | 89 | 13.0±8.5 | 126 | 32.6±15.3 | 111 | 29.9±13.6 |
| $|\mathcal{E}_\mathcal{G}|$ | 48 | 27.0±6.1 | 90 | 62.7±17.9 | 4,996 | 386.1±442.4 | 5868 | 263.7±443.1 | 298 | 124.3±51.0 | 238 | 64.6±29.9 |
| $|\mathcal{X}_\mathcal{G}|$ | 1 | 1±0 | 1 | 1±0 | 1 | 1±0 | 3 | 2.1±0.3 | 1 | 1±0 | 38 | 5.2±4.0 |
| $|\mathcal{Y}_\mathcal{G}|$ | 1 | 1±0 | 1 | 1±0 | 1 | 1±0 | 1 | 1±0 | 1 | 1±0 | 1 | 1±0 |

**Algorithm 2** Benchmarking data construction.

**input** a set of directed graphs $\mathcal{G}_1, \mathcal{G}_2, \cdots, \mathcal{G}_D$, a homogeneous pattern structure $\mathcal{P}'$
1: construct a set of directed homogeneous graphs $\mathcal{G}'_1, \mathcal{G}'_2, \cdots, \mathcal{G}'_D$ where vertex labels and edge labels are dropped
2: conduct subgraph isomorphism counting for pattern $\mathcal{P}'$ over homogeneous graphs $\mathcal{G}'_1, \mathcal{G}'_2, \cdots, \mathcal{G}'_D$
3: create a heterogeneous pattern set $\mathcal{S}$
4: **if** the average counting per graph is less than or equal to 1.0 **then**
5:     get the average vertex label integer $\lceil \overline{x} \rceil$ and average edge label integer $\lceil \overline{y} \rceil$
6:     **for** iter vertex label assignment $\mathcal{X}_\mathcal{P}$ from $\{1, 2, \cdots, \lceil \overline{x} \rceil\}^{|\mathcal{V}_{\mathcal{P}'}|}$ **do**
7:         **for** iter edge label assignment $\mathcal{Y}_\mathcal{P}$ from $\{1, 2, \cdots, \lceil \overline{y} \rceil\}^{|\mathcal{E}_{\mathcal{P}'}|}$ **do**
8:             construct a heterogeneous pattern $\mathcal{P}$ with the same structure of $\mathcal{P}'$, and two label assignments $\mathcal{X}_\mathcal{P}$ and $\mathcal{Y}_\mathcal{P}$
9:             conduct subgraph isomorphism counting for pattern $\mathcal{P}$ over graphs $\mathcal{G}_1, \mathcal{G}_2, \cdots, \mathcal{G}_D$
10:             **if** the average counting per graph is greater than 1.0 **then**
11:                 add current heterogeneous pattern $\mathcal{P}$ to the pattern set $\mathcal{S}$
12:             **end if**
13:         **end for**
14:     **end for**
15: **end if**
**output** the heterogeneous pattern set $\mathcal{S}$

In order to obtain meaningful and challenging predictions, we enumerate all permutations of vertex labels and permutations of edge labels from the 3-stars, triangles, tailed triangles, and chordal cycles. Furthermore, to improve the quality of the data, we have filtered out patterns with an average frequency of less than 1.0 across the entire dataset. The program can be summarized by Algorithm 2. We consistently enumerate the permutations of vertex labels and edge labels, and we check the validity of heterogeneous patterns. The subgraph isomorphism counting is efficiently performed by the VF2 algorithm Cordella et al. (2004a) in parallel. The construction time for each dataset is less than three hours.

## A.2 Parameter Selection

Graph kernels do not require any parameters to be tuned. However, the polynomial kernel and the Gaussian kernel both have hyper-parameters Specifically, for polynomial kernels, we fix the power of the polynomial to 3 and tune the factor of the radix among $\{2e\text{-}5, 2e\text{-}4, \cdots, 1\}$; for Gaussian kernels, we search for the hyper-parameter $2\sigma^2$ in the range of $\{1e\text{-}5, 1e\text{-}4, \cdots, 1e5\}$. When using SVM, we tune the regularization parameter $C$ in the range of $\{1e\text{-}2, \cdots, 1e4\}$; when using Ridge, we tune the regularization parameter $\alpha$ in the range of $\{1e\text{-}4, 1e\text{-}3, \cdots, 1e2\}$. Models are trained and selected based on the validation set, with the mean squared error (MSE) serving as the evaluation metric. The seed is fixed as 2023, and we do not observe the performance change with different seeds.

## B Experimental Results

### B.1 Trivial Baselines and Neural Networks

Two trivial baselines ignore the input data but always make predictions based on the training statistics. For example, the **Zero** baseline always returns zeros because a random graph is highly unlikely to be matched

Table 3: Performance on subgraph isomorphism counting with naive baselines and neural networks.

| Models | Homogeneous | | | | | | | | Heterogeneous | | | |
| | Erdős-Renyi | | Regular | | IMDB-BINARY | | IMDB-MULTI | | ENZYMES | | NCI109 | |
| | RMSE | MAE | RMSE | MAE | RMSE | MAE | RMSE | MAE | RMSE | MAE | RMSE | MAE |
|---|---|---|---|---|---|---|---|---|---|---|---|---|
| Zero | 92.532 | 51.655 | 198.218 | 121.647 | 138262.003 | 37041.171 | 185665.874 | 33063.770 | 64.658 | 25.110 | 16.882 | 7.703 |
| Avg | 121.388 | 131.007 | 156.515 | 127.211 | 133228.554 | 54178.671 | 182717.385 | 53398.301 | 59.589 | 31.577 | 14.997 | 8.622 |
| CNN | 20.386 | 13.316 | 37.192 | 27.268 | 4808.156 | 1570.293 | 4185.090 | 1523.731 | 16.752 | 7.720 | 3.096 | 1.504 |
| LSTM | 14.561 | 9.949 | 14.169 | 10.064 | 10596.339 | 2418.997 | 10755.528 | 1925.363 | 20.211 | 8.841 | 4.467 | 2.234 |
| TXL | 10.861 | 7.105 | 15.263 | 10.721 | 15369.186 | 3170.290 | 19706.248 | 3737.862 | 25.912 | 11.284 | 5.482 | 2.823 |
| RGCN | 9.386 | 5.829 | 14.789 | 9.772 | 46074.355 | 13498.414 | 69485.242 | 12137.598 | 23.715 | 11.407 | 1.217 | 0.622 |
| RGIN | 6.063 | 3.712 | 13.554 | 8.580 | 31058.764 | 6445.103 | 26546.882 | 4508.339 | 8.119 | 3.783 | 0.511 | 0.292 |
| CompGCN | 6.706 | 4.274 | 14.174 | 9.685 | 32145.713 | 8576.071 | 26523.880 | 7745.702 | 14.985 | 6.438 | 1.271 | 0.587 |

by a heterogeneous pattern. The **Avg** baseline tends to predict the average count based on the training data, as the maximum expectation. As shown in Table 3, we can observe that predicting zeros usually yields better absolute errors than predicting the average, indicating the difficulty of the isomorphism counting task. Furthermore, the errors can reach values in the hundreds for both synthetic and real-life data, highlighting the challenge involved.

As a problem of learning to predict, we compare our graph kernel methods with neural networks. Liu et al. (2020) have released an implementation of a neural subgraph isomorphism counting framework[1], which we directly employ to report the results of **CNN** Kim (2014), **LSTM** Hochreiter & Schmidhuber (1997), **TXL** Dai et al. (2019), **RGCN** Schlichtkrull et al. (2018), **RGIN** Liu et al. (2020), and **CompGCN** Vashishth et al. (2020) in Table 3. Graph neural networks (RGCN, RGIN, and CompGCN) outperform sequence models. However, sequence models can still provide relatively accurate predictions when the data is extremely challenging, such as in the case of *IMDB-BINARY* and *IMDB-MULTI*. This encourages researchers to generalize graph neural networks to handle complicated cases, especially considering that RGIN consistently achieves the best results on other datasets.

## B.2 SVM vs. Ridge

We compare the performance of SVM and Ridge (precisely, Kernel Ridge) regression using kernel tricks for subgraph isomorphism counting, which is a common practice to evaluate the performance of these two regressors. The results are shown in Figure 6, and it can be observed that the performance of the two regressors is comparable, with Ridge performing slightly better. There are two main reasons for this. Firstly, Ridge is solved using Cholesky factorization in the closed form, which typically achieves better convergence than the iterative optimization used in SVM. Secondly, the objective of Ridge is to minimize the sum of squared errors, which is more straightforward and suitable for the regression task compared to the $\epsilon$-insensitive loss function used in SVM. Given the superior performance of Ridge, we mainly report the results of Ridge in the following experiments.

In addition to the comparison between SVM and Ridge, we also observed an increase in errors with the polynomial kernel trick. While the number of matched subgraphs is typically small, it can be very large for certain structures such as complete graphs or cliques. The polynomial kernel trick can easily lead to fluctuations due to extreme values, resulting in performance fluctuations and even overflow. Therefore, our focus primarily lies on the original kernels and the Gaussian kernel trick.

## B.3 Normalization

Postprocessing is a common technique used in regression, such as feature normalization to reduce variance and regularization to prevent overfitting. A common way to normalize the Gram matrix involves dividing each element by the square root of the product of the corresponding row and column norms, resulting in $\boldsymbol{K}_{\text{norm}_{i,j}} = \boldsymbol{K}_{i,j} / \sqrt{\boldsymbol{K}_{i,i} \cdot \boldsymbol{K}_{j,j}}$. We investigate the impact of normalization on graph kernels, which are illustrated in Table 4. As expected, normalization significantly harms the performance, regardless of whether the Gaussian kernel is applied or not. We analyze the reason behind this phenomenon using the example

---

[1]https://github.com/HKUST-KnowComp/NeuralSubgraphCounting

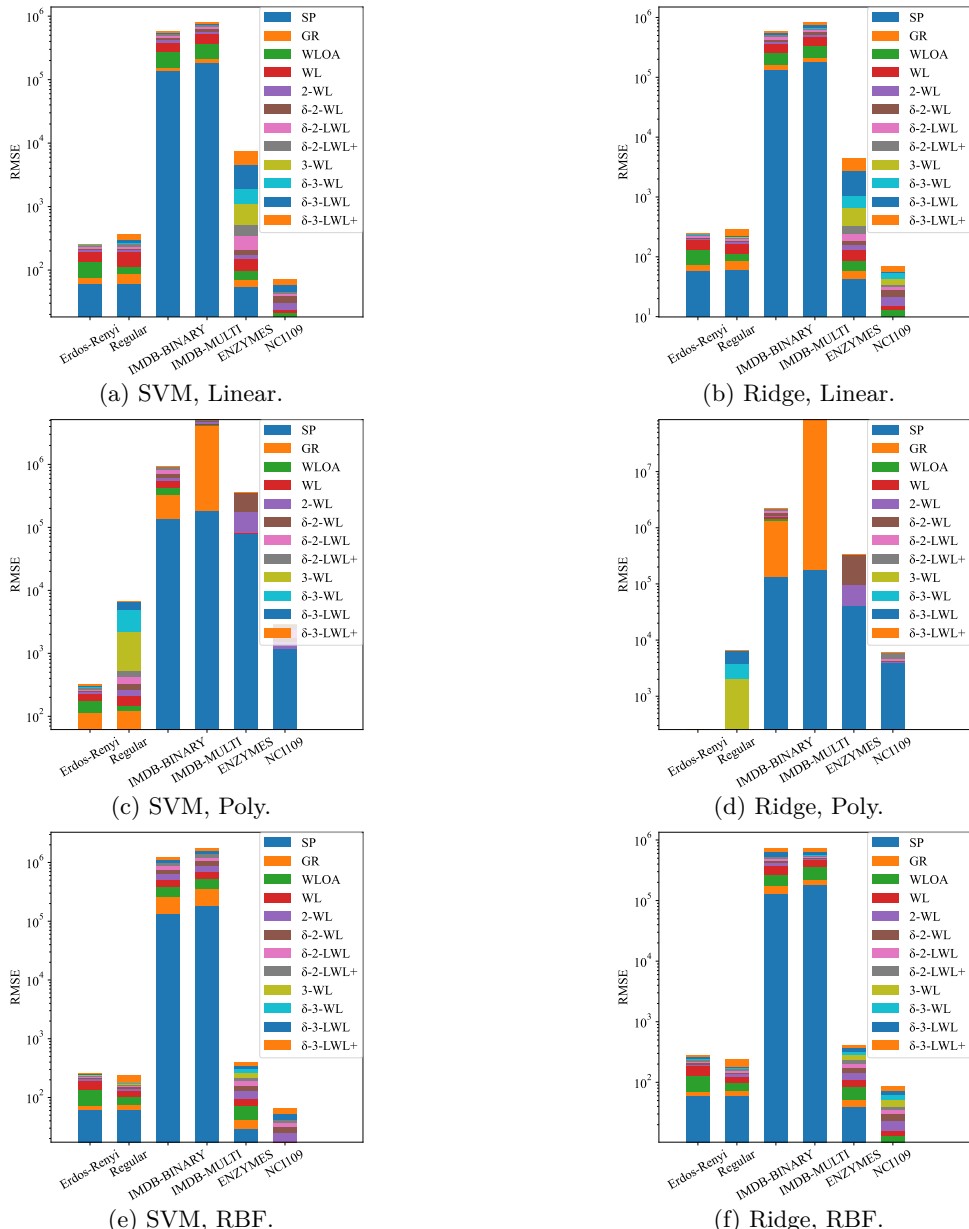

Figure 6: Performance on subgraph isomorphism counting with SVM and Ridge with kernel tricks, where "Poly" and "RBF" indicatet the polynomial and Gaussian kernels, and the out-of-memory ("OOM") and not-a-number ("NaN") are regarded as zeros in the plots.

of the linear kernel. The graph kernel matrix normalization is equivalent to the normalization of the graph representation. Assume the obtained graph representaion is $\boldsymbol{g}_i$, then the corresponding normalized graph representation is $\boldsymbol{g}_{\text{norm}}$. The element of $\boldsymbol{g}_i[j]$ is $k(\mathcal{G}_i, \mathcal{G}_j)$, which is calculated by the inner product of feature vectors. The unnormalized feature vector is equivalent to the concatenation of the pattern histogram. Thus, $k(\mathcal{G}_i, \mathcal{G}_j)$ is the inner product of the histograms of the two graphs, i.e., $\left\langle \left\| \sum_{t=0}^{T} \boldsymbol{C}_{\mathcal{V}_{\mathcal{G}_i}}^{(t)}, \right\| \sum_{t=0}^{T} \boldsymbol{C}_{\mathcal{V}_{\mathcal{G}_j}}^{(t)} \right\rangle$ for WL kernel. The normalized Gram matrix actually calculates the cosine similarity, i.e., $k_{\text{norm}}(\mathcal{G}_i, \mathcal{G}_j) =$

Table 4: Performance comparison with or without normalization.

| Models | Homogeneous | | | | | | | | Heterogeneous | | | |
| | Erdős-Renyi | | Regular | | IMDB-BINARY | | IMDB-MULTI | | ENZYMES | | NCI109 | |
| | RMSE | MAE | RMSE | MAE | RMSE | MAE | RMSE | MAE | RMSE | MAE | RMSE | MAE |
|---|---|---|---|---|---|---|---|---|---|---|---|---|
| **Ridge, Linear** | | | | | | | | | | | | |
| SP | 58.721 | 34.606 | 60.375 | 41.110 | 131672.705 | 56058.593 | 181794.702 | 54604.564 | 43.007 | 14.422 | 4.824 | 2.268 |
| GR | 14.067 | 7.220 | 23.775 | 12.172 | **30527.764** | 7894.568 | **30980.135** | 6054.027 | 14.557 | 5.595 | 5.066 | 2.066 |
| WLOA | 58.719 | 34.635 | 25.905 | 17.003 | 96887.226 | 28849.659 | 117828.698 | 25808.362 | 28.911 | 11.807 | 3.142 | 1.142 |
| WL | 58.719 | 34.635 | 56.045 | 33.891 | 107500.276 | 41523.359 | 147822.358 | 49244.753 | 46.466 | 14.920 | **1.896** | **0.746** |
| 2-WL | 10.452 | 5.561 | 12.353 | 7.906 | 34734.939 | 9161.265 | 47075.541 | 13751.520 | 26.903 | 11.018 | 7.003 | 3.060 |
| δ-2-WL | 9.921 | 4.164 | 8.751 | 5.663 | 33336.019 | 9265.499 | 47075.541 | 13751.520 | 27.528 | 11.286 | 6.910 | 3.039 |
| δ-2-LWL | 11.342 | 4.757 | 11.020 | 7.230 | 38507.321 | 16105.742 | 47075.541 | 13751.520 | 54.915 | 10.079 | 2.605 | 1.072 |
| δ-2-LWL⁺ | 11.132 | 4.687 | 11.795 | 7.703 | 38507.321 | 16105.742 | 47075.541 | 13751.520 | 89.581 | 10.911 | 2.584 | 1.068 |
| 3-WL | **4.096** | **1.833** | 4.038 | 2.330 | OOM | OOM | OOM | OOM | 335.940 | 13.790 | 9.721 | 3.314 |
| δ-3-WL | 4.214 | 1.840 | 4.092 | 2.361 | OOM | OOM | OOM | OOM | 387.816 | 15.573 | 9.712 | 3.290 |
| δ-3-LWL | 5.163 | 1.930 | **3.975** | **2.277** | 43894.672 | 8029.452 | 76218.966 | 9022.754 | 1727.556 | 42.346 | 3.872 | 1.375 |
| δ-3-LWL⁺ | 5.151 | 1.931 | 60.375 | 41.110 | 39237.071 | 7240.730 | 76218.966 | 9022.754 | 1719.251 | 42.626 | 12.488 | 6.501 |
| **Ridge, Linear w/ normalization** | | | | | | | | | | | | |
| SP | 58.721 | 34.606 | 60.375 | 41.110 | 131672.705 | 56058.593 | 181794.702 | 54604.564 | 37.066 | 15.686 | 7.776 | 3.572 |
| GR | 45.922 | 23.308 | 53.449 | 29.348 | 119288.171 | 47374.424 | 181431.698 | 54857.785 | 32.970 | 13.297 | 9.611 | 4.093 |
| WLOA | 58.719 | 34.635 | 25.905 | 17.003 | 96120.928 | 28554.467 | 135879.827 | 31445.034 | 29.247 | 12.023 | 3.594 | 1.295 |
| WL | 58.721 | 34.606 | 60.375 | 41.110 | 131672.705 | 56058.577 | 181794.702 | 54604.564 | 27.789 | 11.801 | 4.581 | 1.861 |
| 2-WL | 10.806 | 5.708 | 12.350 | 7.941 | 87428.241 | 29580.956 | 78615.331 | 16385.534 | 35.913 | 15.344 | 8.639 | 4.250 |
| δ-2-WL | 10.143 | 4.129 | 8.999 | 5.874 | 87894.665 | 31526.343 | 80061.204 | 17979.721 | 36.085 | 15.397 | 8.879 | 4.324 |
| δ-2-LWL | 11.934 | 5.024 | 15.585 | 8.976 | 81849.707 | 28816.597 | 82529.598 | 15951.174 | 27.863 | 11.174 | 5.121 | 2.033 |
| δ-2-LWL⁺ | 11.701 | 4.942 | 15.440 | 8.967 | 81280.195 | 28569.835 | 83319.396 | 17110.823 | 27.868 | 11.183 | 5.128 | 2.037 |
| 3-WL | 4.642 | 2.155 | 11.773 | 7.073 | OOM | OOM | OOM | OOM | 32.908 | 14.018 | OOM | OOM |
| δ-3-WL | 4.745 | 2.207 | 12.000 | 7.147 | OOM | OOM | OOM | OOM | 32.851 | 13.989 | OOM | OOM |
| δ-3-LWL | 4.918 | 2.308 | 14.073 | 8.579 | 80474.058 | 25664.344 | 72950.070 | 14878.074 | 31.425 | 12.998 | 3.872 | 1.613 |
| δ-3-LWL⁺ | 4.917 | 2.306 | 60.375 | 41.110 | 81622.690 | 26328.283 | 72715.240 | 14903.663 | 31.416 | 13.009 | 12.488 | 6.501 |
| **Ridge, RBF** | | | | | | | | | | | | |
| SP | 58.721 | 34.606 | 60.375 | 41.110 | 131672.705 | 56058.593 | 181794.702 | 54604.564 | 38.945 | 14.712 | 5.474 | 2.224 |
| GR | 11.670 | 5.663 | 12.488 | 5.012 | 42387.021 | **5110.985** | 41171.761 | **4831.495** | **12.883** | **5.073** | 4.804 | 1.944 |
| WLOA | 58.719 | 34.635 | 25.906 | 17.002 | 92733.105 | 28242.033 | 137300.092 | 34067.513 | 32.827 | 12.230 | 3.215 | 1.261 |
| WL | 58.719 | 34.635 | 25.905 | 17.003 | 109418.159 | 32350.523 | 112515.690 | 25035.268 | 26.313 | 10.933 | 2.227 | 0.837 |
| 2-WL | 11.010 | 5.926 | 12.618 | 8.317 | 40412.745 | 5351.789 | 21910.109 | 2982.532 | 32.424 | 12.948 | 7.164 | 3.271 |
| δ-2-WL | 10.500 | 4.630 | 9.316 | 6.207 | 40412.745 | 5351.789 | 21910.109 | 2982.532 | 32.518 | 13.045 | 7.409 | 3.287 |
| δ-2-LWL | 11.788 | 5.004 | 8.643 | 5.730 | 40412.745 | 5351.789 | 21910.109 | 2982.532 | 29.560 | 11.878 | 5.010 | 1.806 |
| δ-2-LWL⁺ | 11.659 | 4.936 | 8.495 | 5.634 | 40412.745 | 5351.789 | 21910.109 | 2982.532 | 30.525 | 11.977 | 5.001 | 1.799 |
| 3-WL | 4.949 | 2.568 | 4.631 | 2.783 | OOM | OOM | OOM | OOM | 43.909 | 18.509 | OOM | OOM |
| δ-3-WL | 4.896 | 2.536 | 4.567 | 2.745 | OOM | OOM | OOM | OOM | 43.908 | 18.509 | OOM | OOM |
| δ-3-LWL | 16.720 | 2.980 | 5.356 | 3.149 | 89532.736 | 21918.757 | 91445.323 | 17703.656 | 43.909 | 18.510 | 10.925 | 5.320 |
| δ-3-LWL⁺ | 16.721 | 2.972 | 60.375 | 41.110 | 89532.736 | 21918.757 | 91445.323 | 17703.656 | 43.908 | 18.513 | 12.488 | 6.501 |
| **Ridge, RBF w/ normalization** | | | | | | | | | | | | |
| SP | 58.721 | 34.606 | 60.375 | 41.110 | 131672.705 | 56058.593 | 181794.702 | 54604.564 | 30.476 | 12.663 | 6.093 | 2.703 |
| GR | 48.525 | 20.658 | 41.767 | 19.466 | 112530.277 | 44929.590 | 119919.313 | 28394.513 | 32.496 | 10.665 | 8.561 | 3.470 |
| WLOA | 58.719 | 34.635 | 25.905 | 17.003 | 93888.723 | 28559.126 | 135642.945 | 31846.785 | 28.347 | 11.636 | 3.682 | 1.322 |
| WL | 58.721 | 34.606 | 60.375 | 41.110 | 131112.523 | 54003.979 | 181794.702 | 54604.564 | 27.074 | 11.366 | 4.344 | 1.699 |
| 2-WL | 11.019 | 5.796 | 12.467 | 8.064 | 74732.522 | 20986.727 | 72753.757 | 18058.778 | 29.779 | 12.300 | 7.652 | 3.580 |
| δ-2-WL | 10.382 | 4.248 | 9.203 | 5.988 | 74732.522 | 20986.727 | 72753.757 | 18058.778 | 30.062 | 12.331 | 7.512 | 3.555 |
| δ-2-LWL | 12.112 | 5.137 | 11.271 | 7.170 | 74732.522 | 20986.727 | 79586.593 | 16979.269 | 26.008 | 10.462 | 3.926 | 1.740 |
| δ-2-LWL⁺ | 11.871 | 5.054 | 11.220 | 7.132 | 74732.522 | 20986.727 | 76561.843 | 17602.345 | 25.997 | 10.454 | 3.921 | 1.744 |
| 3-WL | 4.686 | 2.175 | 6.113 | 3.315 | OOM | OOM | OOM | OOM | 30.498 | 12.341 | OOM | OOM |
| δ-3-WL | 4.731 | 2.281 | 5.611 | 3.137 | OOM | OOM | OOM | OOM | 30.886 | 12.448 | OOM | OOM |
| δ-3-LWL | 4.685 | 2.232 | 5.780 | 3.228 | 79781.673 | 26393.124 | 73428.449 | 15311.455 | 28.726 | 11.354 | 3.288 | 1.417 |
| δ-3-LWL⁺ | 4.692 | 2.232 | 60.375 | 41.110 | 79334.836 | 26283.169 | 73217.154 | 15336.018 | 28.728 | 11.352 | 12.488 | 6.501 |

$$\frac{\left\langle \left\| {}^{T}_{t=0} \boldsymbol{C}^{(t)}_{\mathcal{V}_{\mathcal{G}_i}}, \left\| {}^{T}_{t=0} \boldsymbol{C}^{(t)}_{\mathcal{V}_{\mathcal{G}_j}} \right\rangle}{\sqrt{\left\langle \left\| {}^{T}_{t=0} \boldsymbol{C}^{(t)}_{\mathcal{V}_{\mathcal{G}_i}}, \left\| {}^{T}_{t=0} \boldsymbol{C}^{(t)}_{\mathcal{V}_{\mathcal{G}_i}} \right\rangle \cdot \left\langle \left\| {}^{T}_{t=0} \boldsymbol{C}^{(t)}_{\mathcal{V}_{\mathcal{G}_j}}, \left\| {}^{T}_{t=0} \boldsymbol{C}^{(t)}_{\mathcal{V}_{\mathcal{G}_j}} \right\rangle}}}$$ . Therefore, the normalized Gram matrix measures the cosine similarity of two color distributions, rather than the similarity of two graph structures. It is not surprising that the normalized Gram matrix performs worse than the unnormalized one, as the information inside distributions is less than the information inside histograms.

## B.4 Neighborhood Information Extraction

Explicit neighborhood information extraction (NIE) is a crucial component for handling homogeneous data by providing edge colors. As demonstrated in Table 5, incorporating NIE consistently enhances the performance

Table 5: Performance comparison with or without neighborhood information extraction.

| Models | Homogeneous | | | | | | | | Heterogeneous | | | |
| | *Erdős-Renyi* | | *Regular* | | IMDB-BINARY | | IMDB-MULTI | | *ENZYMES* | | *NCI109* | |
| | RMSE | MAE | RMSE | MAE | RMSE | MAE | RMSE | MAE | RMSE | MAE | RMSE | MAE |
|---|---|---|---|---|---|---|---|---|---|---|---|---|
| **Ridge, Linear** | | | | | | | | | | | | |
| SP | 58.721 | 34.606 | 60.375 | 41.110 | 131672.705 | 56058.593 | 181794.702 | 54604.564 | 43.007 | 14.422 | 4.824 | 2.268 |
| GR | 14.067 | 7.220 | 23.775 | 12.172 | 30527.764 | 7894.568 | 30980.135 | 6054.027 | 14.557 | 5.595 | 5.066 | 2.066 |
| WLOA | 58.719 | 34.635 | 25.905 | 17.003 | 96887.226 | 28849.659 | 117828.698 | 25808.362 | 28.911 | 11.807 | 3.142 | 1.142 |
| WL | 58.719 | 34.635 | 56.045 | 33.891 | 107500.276 | 41523.359 | 147822.358 | 49244.753 | 46.466 | 14.920 | 1.896 | 0.746 |
| 2-WL | 10.452 | 5.561 | 12.353 | 7.906 | 34734.939 | 9161.265 | 47075.541 | 13751.520 | 26.903 | 11.018 | 7.003 | 3.060 |
| $\delta$-2-WL | 9.921 | 4.164 | 8.751 | 5.663 | 33336.019 | 9265.499 | 47075.541 | 13751.520 | 27.528 | 11.286 | 6.910 | 3.039 |
| $\delta$-2-LWL | 11.342 | 4.757 | 11.020 | 7.230 | 38507.321 | 16105.742 | 47075.541 | 13751.520 | 54.915 | 10.079 | 2.605 | 1.072 |
| $\delta$-2-LWL$^+$ | 11.132 | 4.687 | 11.795 | 7.703 | 38507.321 | 16105.742 | 47075.541 | 13751.520 | 89.581 | 10.911 | 2.584 | 1.068 |
| 3-WL | **4.096** | **1.833** | 4.038 | 2.330 | OOM | OOM | OOM | OOM | 335.940 | 13.790 | 9.721 | 3.314 |
| $\delta$-3-WL | 4.214 | 1.840 | 4.092 | 2.361 | OOM | OOM | OOM | OOM | 387.816 | 15.573 | 9.712 | 3.290 |
| $\delta$-3-LWL | 5.163 | 1.930 | **3.975** | **2.277** | 43894.672 | 8029.452 | 76218.966 | 9022.754 | 1727.556 | 42.346 | 3.872 | 1.375 |
| $\delta$-3-LWL$^+$ | 5.151 | 1.931 | 60.375 | 41.110 | 39237.071 | 7240.730 | 76218.966 | 9022.754 | 1719.251 | 42.626 | 12.488 | 6.501 |
| **Ridge, Linear w/ NIE** | | | | | | | | | | | | |
| SP | 58.721 | 34.606 | 60.375 | 41.110 | 131672.705 | 56058.593 | 181794.702 | 54604.564 | 43.007 | 14.422 | 4.824 | 2.268 |
| GR | 14.067 | 7.220 | 23.775 | 12.172 | 30527.764 | 7894.568 | 30980.135 | 6054.027 | 14.557 | 5.595 | 5.066 | 2.066 |
| WLOA | 58.719 | 34.635 | 25.905 | 17.003 | 33625.086 | 6009.372 | 20858.288 | 2822.391 | 23.478 | 10.037 | 3.203 | 1.133 |
| WL | 58.719 | 34.635 | 56.045 | 33.891 | 66414.032 | 17502.328 | 70013.402 | 13266.318 | 20.971 | 8.672 | 1.772 | 0.704 |
| 2-WL | 10.452 | 5.561 | 12.353 | 7.906 | 34135.090 | 6275.320 | 47069.352 | 13669.964 | 211.105 | 13.200 | 8.747 | 3.051 |
| $\delta$-2-WL | 9.921 | 4.164 | 8.751 | 5.663 | 14914.025 | 3671.681 | 47069.434 | 13671.226 | 238.306 | 14.007 | 7.369 | 2.954 |
| $\delta$-2-LWL | 11.342 | 4.757 | 11.020 | 7.230 | 26549.602 | 4997.981 | 39932.609 | 10177.426 | 243.690 | 9.925 | **1.259** | **0.539** |
| $\delta$-2-LWL$^+$ | 11.132 | 4.687 | 11.795 | 7.703 | 28183.800 | 5240.118 | 37676.903 | 9930.398 | 97.024 | 7.191 | 1.266 | 0.545 |
| 3-WL | **4.096** | **1.833** | 4.038 | 2.330 | OOM | OOM | OOM | OOM | OOM | OOM | OOM | OOM |
| $\delta$-3-WL | 4.214 | 1.840 | 4.092 | 2.361 | OOM | OOM | OOM | OOM | OOM | OOM | OOM | OOM |
| $\delta$-3-LWL | 5.163 | 1.930 | **3.975** | **2.277** | 1841.533 | 272.143 | 1411.924 | 126.022 | OOM | OOM | OOM | OOM |
| $\delta$-3-LWL$^+$ | 5.151 | 1.931 | 60.375 | 41.110 | 1808.841 | 264.480 | 1346.608 | 123.394 | 380.480 | 19.073 | OOM | OOM |
| **Ridge, RBF** | | | | | | | | | | | | |
| SP | 58.721 | 34.606 | 60.375 | 41.110 | 131672.705 | 56058.593 | 181794.702 | 54604.564 | 38.945 | 14.712 | 5.474 | 2.224 |
| GR | 11.670 | 5.663 | 12.488 | 5.012 | 42387.021 | 5110.985 | 41171.761 | 4831.495 | **12.883** | **5.073** | 4.804 | 1.944 |
| WLOA | 58.719 | 34.635 | 25.906 | 17.002 | 92733.105 | 28242.033 | 137300.092 | 34067.513 | 32.827 | 12.230 | 3.215 | 1.261 |
| WL | 58.719 | 34.635 | 25.905 | 17.003 | 109418.159 | 32350.523 | 112515.690 | 25035.268 | 26.313 | 10.933 | 2.227 | 0.837 |
| 2-WL | 11.010 | 5.926 | 12.618 | 8.317 | 40412.745 | 5351.789 | 21910.109 | 2982.532 | 32.424 | 12.948 | 7.164 | 3.271 |
| $\delta$-2-WL | 10.500 | 4.630 | 9.316 | 6.207 | 40412.745 | 5351.789 | 21910.109 | 2982.532 | 32.518 | 13.045 | 7.409 | 3.287 |
| $\delta$-2-LWL | 11.788 | 5.004 | 8.643 | 5.730 | 40412.745 | 5351.789 | 21910.109 | 2982.532 | 29.560 | 11.878 | 5.010 | 1.806 |
| $\delta$-2-LWL$^+$ | 11.659 | 4.936 | 8.495 | 5.634 | 40412.745 | 5351.789 | 21910.109 | 2982.532 | 30.525 | 11.977 | 5.001 | 1.799 |
| 3-WL | 4.949 | 2.568 | 4.631 | 2.783 | OOM | OOM | OOM | OOM | 43.909 | 18.509 | OOM | OOM |
| $\delta$-3-WL | 4.896 | 2.536 | 4.567 | 2.745 | OOM | OOM | OOM | OOM | 43.908 | 18.509 | OOM | OOM |
| $\delta$-3-LWL | 16.720 | 2.980 | 5.356 | 3.149 | 89532.736 | 21918.757 | 91445.323 | 17703.656 | 43.909 | 18.510 | 10.925 | 5.320 |
| $\delta$-3-LWL$^+$ | 16.721 | 2.972 | 60.375 | 41.110 | 89532.736 | 21918.757 | 91445.323 | 17703.656 | 43.908 | 18.513 | 12.488 | 6.501 |
| **Ridge, RBF w/ NIE** | | | | | | | | | | | | |
| SP | 58.721 | 34.606 | 60.375 | 41.110 | 131672.705 | 56058.593 | 181794.702 | 54604.564 | 38.945 | 14.712 | 5.474 | 2.224 |
| GR | 11.670 | 5.663 | 12.488 | 5.012 | 42387.021 | 5110.985 | 41171.761 | 4831.495 | **12.883** | **5.073** | 4.804 | 1.944 |
| WLOA | 58.719 | 34.635 | 25.906 | 17.002 | 31409.659 | 6644.798 | 19456.664 | 3892.678 | 24.429 | 10.354 | 3.163 | 1.189 |
| WL | 58.719 | 34.635 | 25.905 | 17.003 | 48568.177 | 17533.158 | 71434.770 | 20472.124 | 23.155 | 9.302 | 2.026 | 0.805 |
| 2-WL | 11.010 | 5.926 | 12.618 | 8.317 | 28036.076 | 5266.623 | 48004.143 | 14046.171 | 34.729 | 14.580 | 8.301 | 3.679 |
| $\delta$-2-WL | 10.500 | 4.630 | 9.316 | 6.207 | 15241.302 | 3289.949 | 48004.217 | 14047.425 | 34.707 | 14.584 | 8.266 | 3.669 |
| $\delta$-2-LWL | 11.788 | 5.004 | 8.643 | 5.730 | 25849.115 | 4842.077 | 30846.779 | 6642.524 | 33.838 | 13.947 | 6.620 | 2.807 |
| $\delta$-2-LWL$^+$ | 11.659 | 4.936 | 8.495 | 5.634 | 27368.926 | 5065.269 | 30093.401 | 6593.717 | 33.839 | 13.948 | 6.619 | 2.807 |
| 3-WL | 4.949 | 2.568 | 4.631 | 2.783 | OOM | OOM | OOM | OOM | OOM | OOM | OOM | OOM |
| $\delta$-3-WL | 4.896 | 2.536 | 4.567 | 2.745 | OOM | OOM | OOM | OOM | OOM | OOM | OOM | OOM |
| $\delta$-3-LWL | 16.720 | 2.980 | 5.356 | 3.149 | 856.975 | 160.003 | **833.037** | **75.286** | OOM | OOM | OOM | OOM |
| $\delta$-3-LWL$^+$ | 16.721 | 2.972 | 60.375 | 41.110 | **757.736** | **148.417** | 886.330 | 88.512 | 43.918 | 18.491 | OOM | OOM |

of both linear and RBF kernels. Specifically, the RBF kernel combined with NIE proves to be more effective for homogeneous data, while the linear kernel shows substantial improvement when applied to heterogeneous data. The most significant enhancements are observed on the highly challenging *IMDB-BINARY* and *IMDB-MULTI* datasets, where the RMSE is significantly reduced from 30,527.764 to 757.736 and from 21,910.109 to 833.037, respectively. However, for the remaining two heterogeneous datasets, neighborhood information does not outperform as well as the GR kernel in the *ENZYMES* dataset. Upon analyzing the differences between the GR kernel and our proposed NIE, we find that high-order topologies such as triangles and wedges may be more powerful than first-order topologies. However, it is noteworthy that the 3-WL-family kernels may perform poorly on heterogeneous data. These findings serve as a foundation for further research and advancements in the field of graph kernels.

