# OpenReview forum: "Towards Subgraph Isomorphism Counting with Graph Kernels"
_TMLR — Rejected by TMLR_

### Review · Reviewer_Hyf6 · 2024-08-19

**Summary Of Contributions:**

This paper presents a study of various graph kernels and their extensions with neighbourhood information applied to the subgraph isomorphism counting problem. As claimed by the authors, they are among the first to study this problem from the graph kernel perspective. Experimental results demonstrate benefits of graph kernels -- also when compared against various GNN architectures -- while showing a good outcome for including neighbourhood information, and indicating they are still underperformant in heterogeneous graphs.

**Audience:**

Yes

**Claims And Evidence:**

No

**Requested Changes:**

One very important issue I have with the paper in its present form is that, while it provides a delightful introduction, a lot of the material it presents during its first ~4-5 pages is, indeed, rudimentary material and it does not represent a novel contribution. Additionally, the abstract and introduction are lukewarm at best at describing what the paper's research novelty actually is. As such, I would recommend a rewrite of both the abstract and introduction to very clearly state (ideally in bullet point form in the intro) what are the specific contributions this paper offers. Further, it would make sense to signposts these contributions throughout the introduction---e.g. whenever a generic concept (such as a Gram matrix) is introduced, the Authors could mention why this concept will be important in their proposal.

Secondly, I found that the results presented could have used some confidence estimates -- at least for the models which can be trained from multiple initialisations, such as GNNs. This will allow to ascertain better how the GNNs compare against the kernel approaches. Further, I found that the paper seems to mainly rely on GNNs which are bounded by 1-WL without considering more expressive variants. I find this to be somewhat limiting as a baseline, considering the authors go beyond 1-WL in their kernel models. Some popular suggestions worth considering include:

* 2-3-4-GNNs (Morris et al., AAAI'19)
* PPGNs (Maron et al., NeurIPS'19)
* Random Features GNN (Sato et al., SDM'21)
* CWN (Bodnar et al., NeurIPS'21)
* GSN (Bouritsas et al., TPAMI'22)
* TokenGT (Kim et al., NeurIPS'22)
* Graph Transformers (Müller et al., TMLR'24)

In addition, given that the topic at hand involves subgraph isomorphism, would a subgraph GNN make sense? There's many recent references, e.g. Frasca, Bevilacqua et al. (NeurIPS'22) is a good one.

I'm not saying the work should compare against _all_ of those to be acceptable, but a representative subset of these more powerful GNNs would make a lot of sense to be included in the experiments, and they should probably all be discussed in the related work.

Lastly, I found Figure 5 very hard to draw any conclusions from -- the varied scale of the bar charts and many numbers stacked against each other made it hard to discern the differences in most cases. It would be helpful if this information were summarised in a table -- perhaps in the Appendix if more convenient.

Minor nit: in one place the symbol $h_\mathrm{KL}$ is used; it should be $h_\mathrm{WL}$.

**Strengths And Weaknesses:**

This is an interesting paper that was generally quite easy to read. I am not an expert in graph kernels and the introduction was most educational -- more accessible than most other papers in the field I've previously read.

Given my lack of context on graph kernel literature, I am not able to verify whether the authors' ideas and incorporation of neighbourhood information are novel -- I would defer to the other reviewers to assess that. The ideas are certainly sensible and the results are useful, both for indicating where the current state-of-the-art is and highlighting valuable avenues for future work.

That being said, I felt it was somewhat unclear what the paper's aims and contributions are, the results could do with more understanding of their robustness and significance, and the presentation of the results could be improved. I believe these issues are important to resolve before I can recommend the paper for acceptance. Please see the "Requested Changes" section for an outline.

---

### Review · Reviewer_ix6t · 2024-08-22

**Summary Of Contributions:**

This paper explores the usage of graph kernels in counting subgraph isomorphisms. To be more specific, the paper proposes combining various graph kernels with neighborhood information, and SVM or Ridge. The empirical results show that the proposed method can significantly improve the accuracy of subgraph isomorphism counting.

**Audience:**

Yes

**Broader Impact Concerns:**

No ethical concerns in this paper.

**Claims And Evidence:**

Yes

**Requested Changes:**

I'm not very familiar with the topic of counting subgraph isomorphism. Since the WL algorithm is unable to distinguish all non-isomorphic graphs, is there any theoretical analysis of its error rate? If so, could the theoretical analysis be extended to the algorithm proposed in this paper?

**Strengths And Weaknesses:**

Strengths:
1. This paper explores the usage of graph kernels in counting subgraph isomorphisms. To be more specific, the paper proposes combining various graph kernels with neighborhood information, and SVM or Ridge.
2. The empirical results show that the proposed method can significantly improve the accuracy of subgraph isomorphism counting.

Weakness:
1. Although improved significantly, the accuracy on IMDB datasets is still very low thus makes it unusable.
2. There seems to be no theoretical analysis or guarantees for the error rate.

---

### Review · Reviewer_cHaf · 2024-09-10

**Summary Of Contributions:**

This manuscript looks at the problem of efficiently estimating the number of subgraph isomorphisms.  The primary contribution of the manuscript is to look at using graph kernels to efficiently estimate similarity, which can be effective in reducing the necessary computation and rapidly comparing many graphs.

**Audience:**

Yes

**Broader Impact Concerns:**

None.

**Claims And Evidence:**

No

**Requested Changes:**

The authors need to:

Clarify the utility of the scientific problem and its relationship to the cited literature.

Be explicit about the precise methodological contributions of this manuscript.

Provide necessary details about training of all methods, provide uncertainties, and statistically evaluate performance.

These edits are necessary, and hinder the utility of the manuscript as is.

**Strengths And Weaknesses:**

# Strengths:

The manuscript is a mostly clear description that appropriately covers the background of the topic, and gives detailed mathematical descriptions and algorithms.

# Weaknesses:

To begin with, it is not clear how exactly widespread this challenge or problem is.  Many of the listed references in the beginning to motivate the problem seem to be really about subtype discovery (e.g., Milo et al, 2002; Shen et al, 2019; Kuramochi & Karypis 2004;...) rather than isomorphism counting in particular.  While these are related tasks in some sense, based on the introduction it is unclear to me how scientifically grounded this task actually is. These references need to be clarified as for how they motivate this exact problem.  The scientific motivation needs to be made much clearer, and easily understandable.

While the writing is fairly clear, it is challenging to identify exactly what are the precise methodological contributions of this manuscript.  As is clearly covered, the isomorphism tests have been around for decades.  The idea of graph kernel methods, likewise, have been around for a while, as has the concept of splitting up a larger graph into smaller subtasks.  Likewise, the kernel tricks seem to be relatively standard approaches.  Based on my reading of the manuscript, it seems like the primary machine learning development is that the graph kernels are being applied to subgraphs on a new prediction task.  As such, the contributions really need to be made very explicit so that they are easy to follow and identify.

On the experiments:

Details on how the data was split and why the test/validation numbers are so large are not provided.  As it is atypical to have test sets much larger than the train (or the same size), this needs rationale and details.

Details on how the baseline algorithms/neural networks were tuned is absent in the supplemental material.  It needs to be made explicit, especially given the relatively small performance difference between baseline models and the proposed models.

---

### Decision · Action_Editor_HMnc · 2024-10-23

**Recommendation:** Reject

**Comment:**

This manuscript considers the problem of subgraph isomorphism counting -- a problem known to be quite difficult. The authors leverage graph kernels to approach this problem and present some empirical results.

As submitted, the findings of this paper _might_ be of interest to some in the TMLR community. However, during the review phase, some reviewers expressed concern regarding a lack of clarity in the exact nature of the contributions in this work and its relation to the broader literature. I believe this clarity could potentially be provided via discussion and/or major revisions, but the authors failed to comment during the discussion phase.

Some authors also expressed some concerns regarding the empirical study in the work, including questions regarding the experimental design, the presentation of the results, and the lack of potentially insightful baselines. Again, some of these concerns may have been addressable in discussion.

Given these unaddressed concerns, I am recommending rejection of the submitted manuscript. But the authors could consider resubmission of their work after major revision clarifying the contributions and addressing the concerns with the empirical study,

**Audience:**

Unclear, as the exact contributions of the work appear to be minor and lack clarity as submitted (see other criterion). Unfortunately, the authors declined to comment during the rebuttal phase, leaving these questions open.

**Claims And Evidence:**

Unclear. During the review phase, some reviewers commented that they found the exact claims made by the submission to be unclear / poorly described by the authors. Unfortunately, the authors declined to comment during the rebuttal phase, leaving these questions open.

**Resubmission Of Major Revision:**

The authors may consider submitting a major revision at a later time.